# MambaGesture: Enhancing Co-Speech Gesture Generation with Mamba and Disentangled Multi-Modality Fusion

Chencan Fu*
Zhejiang University
Hangzhou, China
chencan.fu@zju.edu.cn

Yabiao Wang*
Zhejiang University
Hangzhou, China
Tencent Youtu Lab
Shanghai, China
caseywang@tencent.com

Jiangning Zhang†
Zhengkai Jiang
Tencent Youtu Lab
Shanghai, China
vtzhang@tencent.com
zhengkjiang@tencent.com

Xiaofeng Mao
Fudan University
Shanghai, China
xfmao23@m.fudan.edu.cn

Jiafu Wu
Weijian Cao
Tencent Youtu Lab
Shanghai, China
jiafwu@tencent.com
weijiancao@tencent.com

Chengjie Wang
Shanghai Jiao Tong
University
Tencent Youtu Lab
Shanghai, China
jasoncjwang@tencent.com

Yanhao Ge
VIVO
Shanghai, China
halege@vivo.com

Yong Liu†
Zhejiang University
Shanghai, China
yongliu@iipc.zju.edu.cn

## Abstract

Co-speech gesture generation is crucial for producing synchronized and realistic human gestures that accompany speech, enhancing the animation of lifelike avatars in virtual environments. While diffusion models have shown impressive capabilities, current approaches often overlook a wide range of modalities and their interactions, resulting in less dynamic and contextually varied gestures. To address these challenges, we present MambaGesture, a novel framework integrating a Mamba-based attention block, **MambaAttn**, with a multi-modality feature fusion module, **SEAD**. The MambaAttn block combines the sequential data processing strengths of the Mamba model with the contextual richness of attention mechanisms, enhancing the temporal coherence of generated gestures. SEAD adeptly fuses audio, text, style, and emotion modalities, employing disentanglement to deepen the fusion process and yield gestures with greater realism and diversity. Our approach, rigorously evaluated on the multi-modal BEAT dataset, demonstrates significant improvements in Fréchet Gesture Distance (FGD), diversity scores, and beat alignment, achieving state-of-the-art performance in co-speech gesture generation.

## CCS Concepts

• **Human-centered computing** → **Human computer interaction (HCI)**; • **Computing methodologies** → *Motion processing*.

*Both authors contributed equally to this research.
†Corresponding authors.

## Keywords

Gesture Generation, Motion Processing, Data-Driven Animation

**ACM Reference Format:**
Chencan Fu, Yabiao Wang, Jiangning Zhang, Zhengkai Jiang, Xiaofeng Mao, Jiafu Wu, Weijian Cao, Chengjie Wang, Yanhao Ge, and Yong Liu. 2024. MambaGesture: Enhancing Co-Speech Gesture Generation with Mamba and Disentangled Multi-Modality Fusion. In *Proceedings of the 32nd ACM International Conference on Multimedia (MM '24), October 28–November 1, 2024, Melbourne, VIC, Australia.* ACM, New York, NY, USA, 10 pages. https://doi.org/10.1145/3664647.3680625

## 1 Introduction

Co-speech gesture generation, the task of producing human gestures synchronized with audio and other modalities, is crucial for enhancing avatar realism in animation, film, and interactive gaming. Crafting gestures that are both realistic and diverse is a significant challenge and a focal point in contemporary research.

Extensive research has been conducted in this area, leading to the development of numerous innovative approaches. Gesture generation techniques are generally divided into rule-based and data-driven methods, with the latter further categorized into statistical and learning-based approaches. This paper focuses on learning-based co-speech generation methods, which can generally be divided into two categories. 1) Autoencoder-based methods, which employ autoencoders (AEs) [27] or variational autoencoders (VAEs) [26, 48] to translate gesture generation into a reconstruction task, as shown in Figure 1. Despite their computational efficiency, these methods are limited by their architecture, resulting in restricted gesture diversity. 2) Diffusion-based methods, recognized for producing diverse gestures, such as DiffuseStyleGesture+ [46], which combines diffusion models with Transformer encoders and integrates multiple modalities to enhance realism and diversity. However, these methods often fail to fully exploit the rich interactions among multi-modal data, leading to less expressive gestures.

Drawing inspiration from the state space model Mamba, effective in synthetic tasks, language modeling, and audio generation [10], we recognize its potential for co-speech gesture generation. Mamba,

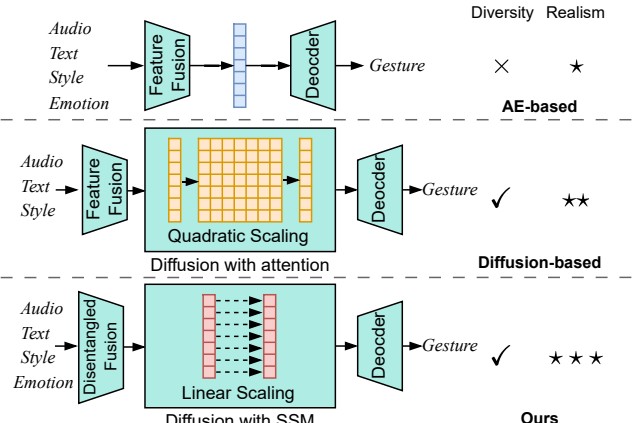

**Figure 1: Comparison of our approach with mainstream co-speech generation methods. (a) Autoencoder (AE)-based methods [26, 27] synthesize gestures by fusing multi-modal data but inherently suffer from limited diversity due to architectural constraints. (b) Diffusion-based methods [44, 46] employ diffusion models with Transformers to generate diverse gestures but are hindered by the quadratic complexity of Transformer and often overlook intricate multi-modal correlations. (c) Our MambaGesture leverages the linear scaling and sequential data processing advantages of the State Space Model to enhance gesture diversity and effectively harness multi-modal data with disentangled feature fusion, ensuring a broader spectrum and higher realism in gesture generation.**

an evolution of RNNs, overcomes their limitations in parallel computation through a parallel scan algorithm and boasts linear scaling, contrasting with the quadratic complexity of traditional transformers. Our exploration into Mamba's application in co-speech gesture generation reveals its capacity to produce gestures that are both realistic and diverse. Experimentally, we find that combining Mamba's sequential modeling strengths with the contextual awareness of attention mechanisms yields the best results. We propose integrating multiple modalities (audio, text, style, and emotion) to significantly elevate the realism and diversity of generated gestures. While audio lays the foundation for gesture cues, adding style and emotion enriches the gestural output, capturing individual expressions and emotional subtleties. Previous research has often focused on audio as the primary input, neglecting the depth of information from other modalities. We argue that audio contains rich details that, when effectively disentangled and combined with text, style, and emotion, can greatly refine gesture synthesis. Our approach aims to fully exploit this rich multi-modal data, with a particular emphasis on audio disentanglement, to achieve more nuanced and contextualized co-speech gesture generation. The concept of our approach is illustrated in Figure 1.

In this work, we introduce MambaGesture, a novel framework that integrates a Mamba-based attention block, MambaAttn, with a multi-modality feature fusion module, SEAD. The MambaAttn block, depicted in Figure 2, leverages the robust sequence modeling capabilities of the state space model Mamba. This integration within

our gesture diffusion model enables the production of gestures that are both realistic and diverse. Additionally, we present the cross-attention enhanced Style and Emotion Aware Disentangled (**SEAD**) feature fusion module. This module introduces a novel technique for disentangling audio to extract personal style and emotion. The combination of MambaAttn's advanced sequence modeling with the SEAD module empowers our framework to generate co-speech gestures that are both diverse and realistic.

Our primary contributions are as follows:

(1) We are the first to introduce the Mamba model to the field of diffusion-based co-speech gesture generation. Mamba's superior sequence modeling capabilities make it well-suited for tasks requiring temporal coherence and dynamic gesture representation.

(2) We introduce the MambaAttn block, which enhances sequential modeling, and the SEAD module, a novel audio disentanglement approach that fuses multi-modal data. The SEAD module effectively captures personal style and emotion from speech audio, enriching the multi-modal information used for gesture generation and leading to more realistic and diverse gestures.

(3) Our extensive experimental evaluation on the large multi-modal BEAT dataset confirms that MambaGesture achieves state-of-the-art performance in co-speech gesture generation. Our method outperforms existing models on several key metrics, highlighting the effectiveness of our contributions.

## 2 Related Work

### 2.1 Co-speech Gesture Generation

Co-speech gesture generation involves producing gestures synchronized with speech audio, a challenging task due to the lack of explicit mappings between speech and gestures.

Gesture generation methodologies can be broadly categorized into rule-based and data-driven approaches. The latter is further subdivided into statistical and learning-based methods[32]. Rule-based methods [4, 19, 20, 29, 35, 40] are known for generating high-quality motions but lack the flexibility and diversity of data-driven systems. Statistical systems [3, 8, 17, 22, 31, 47] model gesture distribution by analyzing the statistics rather than relying on expert-encoded rules, typically involving pre-computing conditional probabilities or assigning prior probability distributions.

Recently, learning-based methods using CNNs, RNNs, and transformers have gained traction. Liu et al. [28] propose a hierarchical approach for gesture generation, considering the hierarchical nature of speech semantics and human gestures. Yoon et al. [49] treat gesture generation as a translation problem, employing a recurrent neural network that utilizes multi-modal contexts of speech text, audio, and speaker identity, incorporating an adversarial scheme to enhance realism. Liu et al. [27] introduce the BEAT dataset, featuring gestures, facial expressions, audio, text, emotions, speaker identity and semantics. Their Cascaded Motion Network (CaMN) synthesizes body and hand gestures using adversarial training across multiple modalities. DisCO [25] disentangles motion into implicit content and rhythm features, feeding the processed features into a motion decoder to synthesize gestures. EMAGE [26] employs Masked Gesture Reconstruction (MG2G) to encode body hints and Audio-Conditioned Gesture Generation (A2G) to decode pre-trained face and body latent features, generating facial and

local body motions using a pre-trained VQ-Decoder. Yi et al. [48] introduce TalkSHOW, which utilizes an autoencoder for facial generation and a VQ-VAE for body and hand motion generation, with an autoregressive model predicting the multinomial distribution of future motion during inference.

These methods often treat co-speech gesture generation as a reconstruction task, facing challenges in establishing mappings between speech and gestures, resulting in limited diversity. However, recent advancements in diffusion-based methods offer a promising alternative, producing gestures with high realism and diversity.

## 2.2 Diffusion-based Gesture Generation

Diffusion models, known for complex data distribution modeling and many-to-many mappings, are gaining popularity for gesture synthesis. Several works have used text as a condition for diffusion models to generate human motion, such as MotionDiffuse [50], FLAME [16], and MDM [38]. Recent research focuses on generating co-speech gestures using diffusion models. Alexanderson [1] adapts the DiffWave architecture [18], replacing dilated convolutions with Transformers or Conformers to enhance performance. Zhu et al. [53] introduce DiffGesture, which concatenates noisy gesture sequences with contextual information in the feature channel for temporal modeling using transformers. DiffuseStyleGesture+ [46] conditions on audio, text, style, and seed gesture, employing an attention-based architecture for denoising. UnifiedGesture [43] utilizes a retargeting network to standardize primal skeletons from various datasets, expanding the data pool and employing VQ-VAE and reinforcement learning for gesture generation refinement. LivelySpeaker [52] emphasizes the importance of semantics in gesture understanding and adopts a two-stage strategy for semantic-aware and rhythm-aware gesture generation. AMUSE [7] disentangles speech into content, emotion, and personal style latent representations, using a motion VAE transformer architecture for the conditional denoising process in latent space. FreeTalker [45] generates spontaneous co-speech gestures from audio and performs text-guided non-spontaneous gesture generation. GestureDiffuCLIP [2] processes text, motion, and video prompts with different CLIP models to achieve style control. DiffSHEG [5] considers expressions as cues for gestures, achieving real-time joint generation of expressions and co-speech gestures.

However, most existing approaches do not consider full modalities, nor do they offer a comprehensive analysis of the interactions between these modalities, potentially leading to less diverse and realistic generated gestures. Our proposed MambaGesture introduces a novel cross-attention enhanced Style and Emotion Aware Disentangled (SEAD) feature fusion module, which cleverly disentangles style and emotion from audio inputs and integrates rich multi-modal conditions (audio, text, style, and emotion) to facilitate the generation of gestures that are both realistic and diverse.

## 2.3 State Space Models

State Space Models (SSMs) [9, 11, 12, 15, 33, 37] have regained popularity in sequence modeling tasks due to their linear or near-linear scaling with sequence length, outperforming attention mechanisms with quadratic scaling. Originating from RNNs and CNNs, SSMs employ state variables to model dynamic systems, making them foundational in fields like control theory and robotics.

The core state space model is represented by the equations:

$$x'(t) = \mathbf{A}x(t) + \mathbf{B}u(t) \qquad (1)$$

$$y(t) = \mathbf{C}x(t) + \mathbf{D}u(t), \qquad (2)$$

where $x(t)$ is an $N$-dimensional latent state, $y(t)$ is the output, $u(t)$ is the input, and $\mathbf{A, B, C, D}$ are system parameters, with $\mathbf{D}u$ often acting as a skip connection.

The Structured State Space Sequence model (S4), introduced by Gu et al.[11], builds on SSMs to achieve generative modeling at scale and fast autoregressive generation. S4 uses the HiPPO matrix to construct parameter $A$, mitigating the challenge of gradient scaling with sequence length. Despite their advantages, SSMs' constant parameters can limit their adaptability and content awareness.

Mamba [10] revolutionizes sequence modeling by maintaining linear complexity, akin to state space models (SSMs), while rivaling Transformers' capabilities. It achieves this through an innovative input-dependent selection mechanism that dynamically adjusts parameters based on the input, significantly enhancing the model's content sensitivity. Additionally, Mamba incorporates computational strategies such as kernel fusion, parallel scan, and recomputation, which collectively streamline the computational process. These features have spurred the creation of Mamba-based applications, including MambaTalk [42], which replaces traditional attention mechanisms with Mamba blocks for efficient and high-quality gesture generation, and Motion Mamba [51], which leverages a U-Net architecture with hierarchical temporal and bidirectional spatial Mamba blocks for advanced denoising capabilities, further augmented by a CLIP text encoder for input conditioning. The development of these applications underscores Mamba's adaptability and its burgeoning role in enhancing motion generation tasks.

## 3 Preliminary

### 3.1 Human Gesture Data Format

Human gestures are predominantly represented using rotation-based formats, with joint rotations typically expressed in SO(3). These can be parameterized through various methods, including Euler angles, axis angles, and quaternions [54].

In this paper, we utilize the BEAT dataset [27], noted for its extensive duration and diverse modalities. The motion capture data in the BEAT dataset is stored in BVH file format, with motion represented via Euler angles: $75 \times 3$ rotations $+ 1 \times 3$ root translation. This dataset includes 27 body joints and 48 hand joints. Consistent with the methodologies employed by DiffuseStyleGesture+ [44], we prefer rotation matrices over Euler angles for joint rotations to enhance the robustness and accuracy of our gesture generation model. Consequently, we treat the root translation as a single joint, resulting in a total of 76 joints. We use all 76 joints for whole-body gesture generation and select 14 upper-body joints, along with the 48 hand joints, for upper-body gestures.

### 3.2 Denoising Diffusion Probabilistic Model

Denoising diffusion probabilistic models (DDPMs) are generative models designed to approximate real-world data distributions, denoted as $q(\mathbf{x}_0)$. Introduced by Ho et al. [14], DDPMs employ a

forward diffusion process that incrementally adds Gaussian noise to data, transitioning it towards a noise distribution, and a reverse process that reconstructs the original data from the noise.

The forward diffusion is a Markov chain described as:

$$q(\boldsymbol{x}_{1:T}|\boldsymbol{x}_0) = \prod_{t=1}^{T} q(\boldsymbol{x}_t|\boldsymbol{x}_{t-1}), \tag{3}$$

$$q(\boldsymbol{x}_t|\boldsymbol{x}_{t-1}) = \mathcal{N}(\boldsymbol{x}_t; \sqrt{1-\beta_t}\boldsymbol{x}_{t-1}, \beta_t \boldsymbol{I}), \tag{4}$$

where $\beta_t$ increases over time, making the data resemble Gaussian noise $\mathcal{N}(\mathbf{0}, \boldsymbol{I})$.

The reverse process reconstructs the data distribution $p_\theta(\boldsymbol{x}_{0:T})$:

$$p_\theta(\boldsymbol{x}_{0:T}) = p_\theta(\boldsymbol{x}_T) \prod_{t=1}^{T} p_\theta(\boldsymbol{x}_{t-1}|\boldsymbol{x}_t), \tag{5}$$

$$p_\theta(\boldsymbol{x}_{t-1}|\boldsymbol{x}_t) = \mathcal{N}(\boldsymbol{x}_{t-1}; \mu_\theta(\boldsymbol{x}_t, t), \Sigma_\theta(\boldsymbol{x}_t, t)), \tag{6}$$

with $\Sigma_\theta(\boldsymbol{x}_t, t)$ as a time-dependent constant. The model approximates the mean of the Gaussian distribution during the reverse process.

Our approach diverges from predicting noise at each step $t$. Instead, we predict the clean data sample $\boldsymbol{x}_0$ directly, following recent methodologies [34, 38, 44], to enhance the generative model's efficiency and accuracy.

## 4 Method

Our approach is structured around two pivotal components: the cross-attention enhanced **S**tyle and **E**motion **A**ware **D**isentangled (**SEAD**) feature fusion module and the **MambaAttn**-based denoising network. At the core of our motion generation lies the diffusion model framework, which employs an iterative process of diffusion (adding noise) and denoising to reconstruct original gestures from a noise distribution, conditioned on audio and additional multi-modal data inputs. The SEAD module is tasked with the intricate fusion of multi-modality data, while the MambaAttn-based denoising network is dedicated to the accurate prediction of gestures. The overview of our proposed MambaGesture is illustrated in Figure 2.

### 4.1 Disentangled Multi-Modal Fusion

Our methodology introduces a progressive series of multi-modality feature fusion techniques. These methods incrementally integrate features from various modalities, evolving from simple concatenation to sophisticated, disentangled fusion.

**SA and SEA Fusion.** Adopting the state-of-the-art DiffuseStyleGesture+ (DSG+) [44, 46] as our baseline, we refine the gesture generation process by conditioning on a set of modalities, including timestep $t$, noisy gesture $x_t$, and conditions $c$. In contrast to DSG+, which employs audio $a$, text $text$, style $s$, and seed gesture $d$, our approach introduces emotion $e$ as a condition while dispensing with the seed gesture. This decision is informed by the recognition that emotion plays a critical role in the natural variation of gestures.

We process each modality through a series of steps to prepare the features for fusion. The timestep is encoded through position encoding and an MLP to produce the time feature $f_t$. The noisy gesture $x_t$ is encoded by a linear layer to obtain the noisy gesture feature $f_g$. Audio features are extracted and enriched with pretrained models to form $f_a$, while text and style features are similarly processed to

yield $f_{text}$ and $f_s$, respectively. Emotion features are encoded to produce $f_e$, with both style and emotion features subject to random masking during training.

The preliminary feature processing of the multi-modality data is completed before training. These preprocessed features are then utilized in the subsequent fusion module.

Previous works have explored multi-modality fusion for input conditions [27, 44]. They perform fusion in a cascaded way or leverage the powerful modeling capabilities of attention mechanism [39]. However, in current diffusion-based gesture generation methods, the process of multi-modality feature fusion is often complex and lacks clarity. We first introduce the **S**tyle **A**ware (**SA**) feature fusion module, which simply uses audio feature $f_a$, text feature $f_{text}$ and style feature $f_s$ as conditions, and concatenates them with the noisy gesture feature $f_g$ and time feature $f_t$. After concatenating all modality features, we employ a cross-local attention [36] module to capture local information within the concatenated feature, resulting in the fused multi-modality feature $f_{fuse}$.

Moreover, many studies overlook the role of emotion. However, emotion can significantly influence our gestures when we speak. To address this, we propose the **S**tyle and **E**motion **A**ware (**SEA**) feature fusion module, building upon SA, which introduces emotion as a generation condition. The emotion feature $f_e$ is fused through concatenation in the same way. By utilizing audio, text, style and emotion, the comprehensive conditions provide a more specific description and command, which benefits our gesture generation.

**SEAD Fusion.** Existing works on multimodal fusion often fail to fully exploit the inherent relationships between modalities. Our **S**tyle and **E**motion **A**ware **D**isentangled (**SEAD-basic**) module, an extension of the SEA module, disentangles style and emotion from audio input using self-supervised learning. It avoids the complexity of two-stage disentanglement methods. This module consists of three independent units that extract style and emotion features from the audio feature $f_a$ as follows:

$$
\begin{aligned}
f_a^s &= \text{Linear}(f_a) \\
f_a^e &= \text{Linear}(f_a) \\
f_a^g &= \text{Linear}(f_a),
\end{aligned}
\tag{7}
$$

where linear layers are used to extract style and emotion features from $f_a$. The extracted $f_a^s$ and $f_a^e$ are aligned with the corresponding style and emotion features using style loss $\mathcal{L}_s$ and emotion loss $\mathcal{L}_e$ to facilitate the separation of personal style and emotion information from the audio.

After disentangling the audio style feature $f_a^s$ and audio emotion feature $f_a^e$, we fuse them with the original style and emotion features to obtain enhanced representations $f_s^h$ and $f_e^h$:

$$f_s^h = \text{Linear}(\text{Cat}(f_a^s, f_s)) \tag{8}$$

$$f_e^h = \text{Linear}(\text{Cat}(f_a^e, f_e)), \tag{9}$$

where Linear denotes a linear layer, and Cat denotes concatenation. By combining them with the original features, we decouple style and emotion from the original audio and enhance them. The remaining feature $f_a^g$ is directly related to the gesture. The remaining process of SEAD-basic is the same as SEA, where we concatenate $f_a^g, f_s^h, f_e^h, f_{text}$ with $f_t$ and $f_g$, and use cross-local attention to obtain the fused feature $f_{fuse}$.

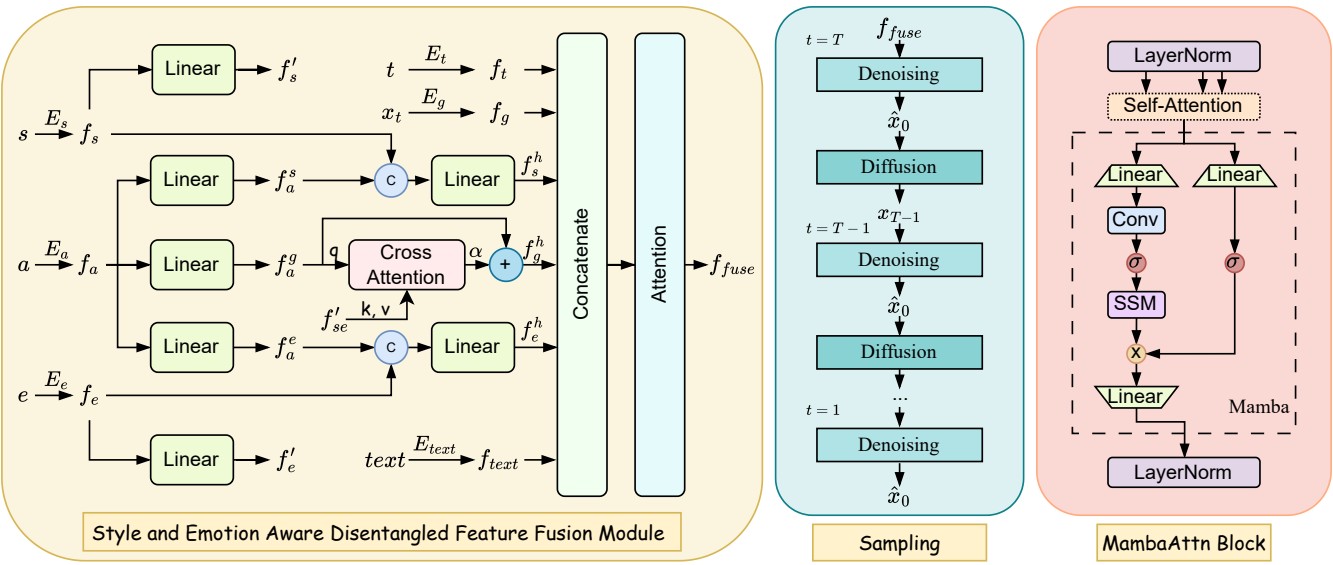

**Figure 2: Overview of our proposed MambaGesture. We introduce a novel feature fusion strategy: the cross-attention enhanced Style and Emotion Aware Disentangled (SEAD) feature fusion module. This module employs style $s$, audio $a$, emotion $e$, and text $text$ as conditions to provide comprehensive information and effectively disentangle style and emotion from the audio. The $f'_{se}$ is obtained by concatenating $f'_s$ and $f'_e$, and projected to original dimension by linear layer. Besides, we present a Mamba-based component termed the *MambaAttn* block, which merges Mamba with its sequence modeling proficiency and employs an attention mechanism to learn global information. Our denoising architecture, MambaAttn denoiser, is composed of a stack of *MambaAttn* blocks and a linear layer. During the sampling phase, we predict the gesture $\hat{x}_0$ by applying the fused conditions within a cyclical denoising and diffusion procedure.**

Building on the SEAD-basic module, the cross-attention enhanced **S**tyle and **E**motion **A**ware **D**isentangled (**SEAD**) feature fusion module further enhances the audio feature $f_a^g$ with the fused style and emotion feature $f'_{se}$ using a cross-attention mechanism:

$$attn = \text{Attention}(Q, K, V) = \text{softmax}(\frac{QK^T}{\sqrt{d}})V, \quad (10)$$

where $Q$ represents the feature $f_a^g$, $K$ and $V$ represent the fused style and emotion feature $f'_{se}$, and $d$ is the dimension of the feature.

The fused style and emotion feature $f'_{se}$ is obtained as follows:

$$f'_{se} = \text{Linear}(\text{Cat}(f'_s, f'_e)). \quad (11)$$

The SEAD module represents the pinnacle of our fusion approach, where the audio feature $f_a^g$ is further refined through cross-attention with fused style and emotion feature $f'_{se}$. This mechanism effectively integrates the audio with the style and emotion, enhancing the representational capability of the audio feature and resulting in a comprehensive fused multi-modality feature $f_{fuse}$.

## 4.2 MambaAttn Denoiser

After integrating the multi-modal conditions into a comprehensive feature representation $f_{fuse}$, we move to the gesture prediction phase. Here, we employ our innovative MambaAttn-based network, MambaAttn Denoiser, to predict gestures $\hat{x}_0$ from the noisy gesture input $x_t$. The MambaAttn Denoiser comprises 8 MambaAttn blocks and a linear projection layer.

**MambaAttn Block.** The MambaAttn block is a novel architectural component that combines the sequential data processing strengths of the Mamba model with the contextual awareness of the attention mechanism. This fusion creates a powerful tool for capturing the intricacies of co-speech gesture generation.

The structure of the MambaAttn block, as shown in Figure 2, includes a self-attention layer, a Mamba block, and two instances of layer normalization. The process begins with the fused feature $f_{fuse}$ undergoing layer normalization, which then enters the self-attention module:

$$Attn = \text{Attention}(Q, K, V) \quad (12)$$

where $Q, K, V = \text{LN}(f_{fuse})$, and LN denotes layer normalization. The self-attention module distills global contextual information, which is then refined by the Mamba block to enhance sequence modeling. The output is subsequently processed as follows:

$$\hat{x}_0 = \text{LN}(\text{Mamba}(Attn)). \quad (13)$$

The final output of the MambaAttn blocks is then projected back to the original dimension of the gesture data through the linear layer, yielding the predicted gesture $\hat{x}_0$.

**Training and Sampling.** To train our networks, we use the Huber loss function as the primary metric for gesture loss $\mathcal{L}_g$:

$$\mathcal{L}_g = E_{x_0 \in q(x_0|c), t \sim [1,T]}[\text{HuberLoss}(x_0 - \hat{x}_0)]. \quad (14)$$

For the style and emotion losses, $\mathcal{L}_s$ and $\mathcal{L}_e$, we use the L1 Loss, formulated as:

$$\mathcal{L}_s = |f_a^s - f_s|$$
$$\mathcal{L}_e = |f_a^e - f_e|. \tag{15}$$

The final loss function is a composite of the gesture, style, and emotion losses, weighted and summed as follows:

$$\mathcal{L} = \mathcal{L}_g + \alpha \mathcal{L}_s + \beta \mathcal{L}_e, \tag{16}$$

where we set $\alpha = 1$ and $\beta = 1$ for simplicity.

Following the DDPM denoising paradigm, we iteratively predict the gesture $\hat{x}_0$ at each timestep $t$, as illustrated in Figure 2, refining our model's ability to generate accurate and lifelike gestures.

## 5 Experiments

### 5.1 Experiment Settings

**Dataset.** We evaluate our method using the BEAT dataset [27], which includes human motions captured at 120Hz via a motion capture system. This dataset features extended conversation audios (approximately 10 minutes each) and brief self-talk audios (around 1 minute each) from 30 diverse speakers. It is multi-modal, offering motion, audio, text, style (identity), emotion, and facial expression annotations. The dataset covers 8 emotions: neutral, anger, happiness, fear, disgust, sadness, contempt, and surprise. Following [41], we select one hour of audio per speaker, splitting the data into 70% for training, 10% for validation, and 20% for testing. We generate both upper body and whole body gestures for comparison with state-of-the-art methods. In ablation studies, we focus on whole body gesture generation.

**Evaluation Metrics.** We use multiple metrics to rigorously assess the quality and diversity of the generated gestures. The Fréchet Gesture Distance (FGD) [49] measures the Fréchet distance between the feature distributions of real and synthesized gestures, similar to the FID [13] used in image generation. The gesture feature extractor, trained unsupervisedly with a reconstruction loss using L1 loss, serves as the basis for this comparison.

To quantify gesture diversity, we use the Diversity Score [21] and L1 Diversity Score [23]. The Diversity Score measures the average feature distance by randomly selecting 500 features and computing the average L1 distance between them.

We also incorporate Semantic-Relevant Gesture Recall (SRGR) [27] and BeatAlign [24] to evaluate the semantic relevance and synchrony of the generated gestures. SRGR, an evolution of the Probability of Correct Keypoint (PCK), measures the semantic relevance of gestures to the accompanying speech. BeatAlign assesses the temporal alignment between audio and gesture beats using Chamfer Distance, providing insight into the rhythmic harmony of the generated gestures with the spoken content.

**Implementation Details.** We downsample motion data from 120 Hz to 30 Hz and segment it into 300-frame clips (10 seconds each). Audio data is downsampled to 16 kHz from a higher sampling rate. We compute a comprehensive set of audio features, including MFCCs, Mel spectrogram features, prosodic features, and pitch onset points (onsets). These features are concatenated with those extracted by the pretrained WavLM Large model [6] to form a rich audio feature set $f_a$. For text data, we employ the pretrained fastText model [30] to extract word vectors from the speech transcripts, which are then processed through a linear layer to obtain the text feature $f_{text}$. Personal style $s$ and emotion $e$ are encoded as one-hot vectors and transformed into corresponding features $f_s$ and $f_e$ via linear layers. The diffusion process is set to 1000 steps. We train each model for 40,000 steps with a batch size of 400, using the AdamW optimizer with a learning rate of $3 \times 10^{-5}$. All experiments are conducted on a single NVIDIA H800 GPU, ensuring reproducibility on standard hardware configurations.

We compare our proposed model with state-of-the-art gesture generation methods, including DiffuseStyleGesture+ (DSG+) [44], CaMN [27], and MDM [38]. We retrain these methods on our partitioned dataset to ensure a fair comparison. CaMN is retrained with audio, text, emotion, and speaker identity as conditions, while DSG+ employs seed gestures, audio, and speaker identity. MDM is conditioned solely on audio features. Our evaluation includes both whole-body and upper-body gesture generation.

### 5.2 Quantitative Results.

The results, summarized in Table 1, demonstrate that our MambaGesture framework achieves the lowest FGD Score for both upper body and whole body gesture generation, indicating a high similarity between the generated gestures and the real data distribution. Specifically, our whole body gesture generation records an FGD Score of 22.11, a significant improvement over MDM's 106.56 and DSG+'s 103.15. Additionally, our method excels in Diversity Score and L1 Diversity Score, with upper-body gestures scoring 374.08 and 875.06, respectively, and whole-body gestures scoring 434.94 and 1128.79, respectively. These scores underscore the enhanced diversity of our generated gestures. Notably, diffusion-based methods such as MDM and DSG+ exhibit better diversity than CaMN, which relies on autoencoders for gesture generation. Although our method does not achieve the highest SRGR Scores, it remains competitive with CaMN. Our MambaGesture excels in BeatAlign for both upper and whole-body gestures, reflecting a more synchronized audio-gesture alignment.

### 5.3 Qualitative Results.

**User Study.** We conduct a user study to subjectively evaluate our proposed method against state-of-the-art methods. Fifteen participants were asked to evaluate 30 gesture samples, each containing gestures generated by four different methods using the same corresponding audio. They were then asked to select the best gesture for each of the following criteria: motion naturalness, smoothness, diversity, and semantic preservation. The preferred method for each criterion was determined based on the number of selections received, and the results are presented as percentages in Table 2.

**Visualization Results.** Figure 3 visualizes the experimental results for the speech transcript "... when you have to work Monday through Friday the whole week, you are very tired ...", a sentence that should elicit rich body movements. Visual comparison shows that gestures generated by CaMN and DSG+ exhibit limited motion, while our approach demonstrates rich motion, highlighting its effectiveness.

Both numerical and visual results corroborate that our method generates realistic and diverse co-speech gestures, advancing the state-of-the-art in the field.

**Table 1: Quantitative comparison of our proposed method with current leading approaches on the BEAT dataset. Bold indicates the top-performing method and underline signifies the second-best performance across various evaluation criteria.**

|            | Method      | FGD Score↓ | Diversity Score↑ | L1Div Score↑ | SRGR Score↑ | BeatAlign↑ |
|------------|-------------|------------|------------------|--------------|-------------|------------|
|            | GT          | -          | 403.27           | 754.75       | -           | 0.894      |
|            | CaMN [27]   | 60.67      | 295.62           | 519.53       | **0.216**   | 0.823      |
| Upper Body | MDM [38]    | 54.13 | 327.82         | 821.18 | 0.208      | 0.823      |
|            | DSG+ [46]   | 60.50      | 358.62    | 748.42       | 0.213 | 0.850 |
|            | Ours        | **32.45**  | **374.08**       | **875.06**   | 0.213 | **0.863**  |
|            | GT          | -          | 395.20           | 850.51       | -           | 0.893      |
|            | CaMN [27]   | 65.74 | 277.06         | 587.12       | **0.241**   | 0.819      |
| Whole Body | MDM [38]    | 106.56     | 331.53           | 1001.52 | 0.229     | 0.810      |
|            | DSG+ [46]   | 103.15     | 352.31    | 789.83       | 0.238 | 0.841 |
|            | Ours        | **22.11**  | **434.94**       | **1128.79**  | 0.237       | **0.853**  |

**Table 2: User Study Results**

| Method    | Natural    | Smooth     | Diversity  | Semantic   |
|-----------|------------|------------|------------|------------|
| CaMN[27]  | 9.78%      | 8.00%      | 3.11%      | 5.78%      |
| MDM[38]   | 7.78%      | 9.11%      | 2.67%      | 5.56%      |
| DSG+[46]  | 21.33%     | 22.22%     | 38.89%     | 21.56%     |
| Ours      | **61.11**% | **60.67**% | **55.33**% | **67.11**% |

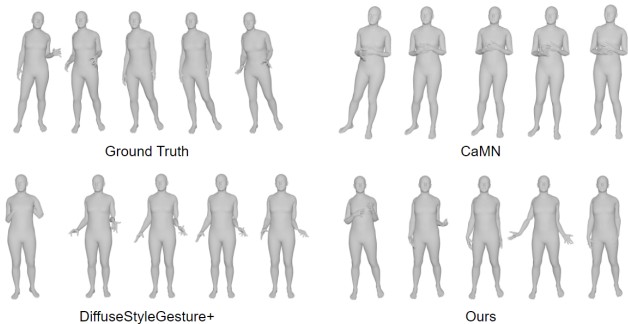

Ground Truth · CaMN · DiffuseStyleGesture+ · Ours

**Figure 3: Visualization results comparing state-of-the-art methods. Speech transcript:** *"... when you have to work Monday through Friday the whole week, you are very tired ..."*

## 5.4 Ablation Studies

To rigorously evaluate the contributions of our proposed method's components, we conduct a series of ablation studies, systematically isolating and analyzing the impact of each module.

**Effectiveness of Proposed Components.** We establish DiffuseStyleGesture+ (DSG+) as our baseline model and incrementally integrate our novel components: SEA, SEAD and the MambaAttn denoiser. The results, detailed in Table 3, show that replacing DSG+'s transformer encoder with our MambaAttn denoiser significantly reduces the Fréchet Gesture Distance (FGD) Score, indicating a closer match to the ground truth gestures. This is accompanied by substantial improvements in both the Diversity Score and the L1 Diversity Score. Although the Semantic-Relevant Gesture Recall (SRGR) Score sees a marginal decrease, the BeatAlign Score improves, underscoring the MambaAttn block's robust capability for

gesture generation. The integration of our SEA module, which introduces emotion as a novel condition, leads to an additional reduction in the FGD Score, with other metrics showing slight variations. The culmination of our fusion approach, the SEAD module, which disentangles style and emotion from audio features and enhances them through cross-attention, further decreases the FGD Score and notably increases the Diversity Score and L1 Diversity Score. These results robustly validate the efficacy of our proposed method.

**Optimal Number of Layers in MambaAttn Denoiser.** An additional ablation study examines the optimal number of layers in the MambaAttn denoiser. Testing configurations of 1, 2, 4, 8, and 12 layers, we find that an 8-layer MambaAttn denoiser yields the best performance across key evaluation metrics, including the FGD Score, Diversity Score, and L1 Diversity Score. Notably, increasing the number of layers to 12 does not confer additional benefits and instead leads to decreased performance. Thus, we select an 8-layer configuration for the MambaAttn denoiser in our final model, as demonstrated by the upper part of Table 4. This finding underscores the importance of balancing model complexity with performance, as overly complex models may suffer from diminishing returns or even performance degradation.

**Design Choices for MambaAttn Block.** Further experimentation is conducted to refine the architectural design of the MambaAttn block. Initially, we consider incorporating a convolutional layer to capture local information, with self-attention to capture global information, while relying on Mamba for sequential modeling. Our experiments with various combinations of these modules, as shown in the lower part of Table 4, indicate that the inclusion of a convolutional layer does not enhance performance. Removing either the self-attention or Mamba components from the block resulted in a significant decrease in the FGD Score, with the removal of Mamba leading to a more pronounced drop in Diversity Score and L1 Diversity Score. Especially when we remove self-attention, with only Mamba and layer norms in our blocks, it can be seen that the Mamba-only architecture also works well for generating realistic and diverse gestures. This suggests that Mamba's sequential modeling ability is crucial for generating diverse gestures. The experiments further confirm this observation.

**Feature Fusion Module Designs.** Lastly, we evaluate various designs for the feature fusion module. The results are presented

**Table 3: Ablation study assessing the contribution of each innovative component within our gesture generation framework. This study systematically alters individual elements to evaluate their impact on the overall performance on the BEAT dataset.**

| No. | Name | FGD Score↓ | Diversity↑ | L1Div Score↑ | SRGR ↑ | BeatAlign↑ |
|-----|------|-----------|-----------|-------------|--------|-----------|
| 1 | Basic DSG+ | 103.15 | 352.31 | 789.83 | **0.238** | 0.841 |
| 2 | + MambaAttn | 64.84 | 389.77 | 1081.14 | 0.233 | **0.855** |
| 3 | + SEA | 29.95 | 387.29 | 955.75 | 0.237 | 0.840 |
| 4 | + SEAD | **22.11** | **434.94** | **1128.79** | 0.237 | 0.853 |

**Table 4: Ablation study examining the influence of the number of layers in the MambaAttn denoiser and the architectural design of the MambaAttn block. This experiment explores the optimal configuration for our model on the BEAT dataset.**

| | No | Name | FGD Score↓ | Diversity Score↑ | L1Div Score↑ | SRGR Score↑ | BeatAlign↑ |
|---|-----|------|-----------|-----------------|-------------|------------|-----------|
| layer number | A1 | MambaAttn-1 | 62.54 | 333.29 | 743.76 | **0.240** | 0.831 |
| | A2 | MambaAttn-2 | 41.17 | 329.03 | 749.73 | **0.240** | 0.851 |
| | A3 | MambaAttn-4 | 36.47 | 343.49 | 895.69 | 0.238 | 0.849 |
| | A4 | MambaAttn-8 | **22.11** | **434.94** | **1128.79** | 0.237 | 0.853 |
| | A5 | MambaAttn-12 | 38.36 | 424.39 | 1110.82 | 0.235 | **0.865** |
| block design | B1 | MambaAttn | **22.11** | **434.94** | **1128.79** | 0.237 | 0.853 |
| | B2 | w/ Conv | 29.88 | 388.81 | 862.08 | 0.238 | 0.861 |
| | B3 | w/o Attn | 46.58 | 358.49 | 864.19 | 0.238 | **0.867** |
| | B4 | w/o Mamba | 44.21 | 329.12 | 771.29 | **0.240** | 0.838 |
| | B5 | w/ Conv, w/o Attn | 93.25 | 418.94 | 895.71 | 0.237 | 0.845 |
| | B6 | w/ Conv, w/o Mamba | 34.10 | 363.65 | 813.65 | 0.239 | 0.858 |
| | B7 | w/ Conv, w/o Attn & Mamba | 75.37 | 324.37 | 760.02 | **0.240** | 0.846 |

**Table 5: Ablation study exploring the effectiveness of different feature fusion strategies in our gesture generation model. The study compares various approaches to integrating multi-modal data, including audio, text, style, and emotion, to determine their impact on the quality of generated gestures on the BEAT dataset.**

| No | Name | FGD Score↓ | Diversity Score↑ | L1Div Score↑ | SRGR Score↑ | BeatAlign↑ |
|----|------|-----------|-----------------|-------------|------------|-----------|
| 1 | Origin DSG+ Input | 64.84 | 389.77 | 1081.14 | 0.233 | **0.855** |
| 2 | Simplify | 32.08 | 387.72 | 968.36 | **0.237** | 0.838 |
| 3 | Simplify+Emo (SEA) | 29.95 | 389.52 | 955.75 | **0.237** | 0.848 |
| 4 | SEA+Disentanglement | 26.61 | 395.51 | 986.89 | **0.237** | 0.850 |
| 5 | SEAD | **22.11** | **434.94** | **1128.79** | **0.237** | 0.853 |

in Table 5. Starting with a simplified fusion approach that directly concatenates audio, text, and style features (our SA fusion module), we observe a substantial decrease in the FGD Score, albeit with a decline in diversity metrics, as this method forgoes the baseline's feature fusion strategy. By adding the emotion modality, we create our SEA module, which further improves the FGD Score. The subsequent disentanglement in our SEAD-basic module results in improvements across the FGD Score, Diversity Score, and L1 Diversity Score. The final enhancement with the fused style and emotion feature $f'_{se}$ in our SEAD module significantly increases both the Diversity Score and L1 Diversity Score.

The detailed results of these ablation studies demonstrate the incremental benefits of each proposed component in our gesture generation framework, providing clear evidence of their individual and collective impact on the model's performance.

## 6 Conclusion

This paper introduces MambaGesture, a novel framework for co-speech gesture generation that leverages the state space model Mamba and a disentangled multi-modality fusion technique. Our approach integrates the MambaAttn block, which combines Mamba's strengths in sequential data processing with the contextual understanding provided by attention mechanisms. Additionally, the SEAD module effectively disentangles style and emotion from audio features, enabling the generation of more realistic and expressive gestures. Comprehensive experiments on the BEAT dataset demonstrate that MambaGesture outperforms state-of-the-art methods across multiple metrics, validating the effectiveness of the SEAD module and MambaAttn block. Future work will address current limitations, such as the slow synthesis speed, and may explore integrating facial expressions to synthesize a complete avatar.

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
