# OpenReview forum: "MambaGesture: Enhancing Co-Speech Gesture Generation with Mamba and Disentangled Multi-Modality Fusion"
_acmmm.org/ACMMM/2024/Conference — MM2024 Poster_

### Official Review · Reviewer_jZQ4 · 2024-05-21

**Rating:** 4
**Confidence:** 4

**Summary:**

"MambaGesture: Enhancing Co-Speech Gesture Generation with Mamba and Disentangled Multi-Modality Fusion" introduces a new framework for generating human gestures synchronized with speech. This framework, MambaGesture, incorporates a Mamba-based attention block (MambaAttn) and a multi-modality feature fusion module (SEAD) to enhance the realism and diversity of co-speech gestures. By leveraging the strengths of Mamba's sequential data processing and the contextual richness of attention mechanisms, the authors propose a model that significantly improves the temporal coherence and contextual relevance of generated gestures.

* MambaAttn Block: Integration of the Mamba model with attention mechanisms to enhance temporal coherence and sequence modeling for gesture generation.
* SEAD Module: Introduction of a Style and Emotion Aware Disentangled feature fusion module that effectively integrates audio, text, style, and emotion modalities to produce realistic and diverse gestures.
* Improved Metrics: Demonstration of state-of-the-art performance on the BEAT dataset, with significant improvements in Fréchet Gesture Distance (FGD), diversity scores, and beat alignment.
* Ablation Studies: Extensive ablation studies that validate the efficacy of each component within the proposed framework, highlighting the contributions of MambaAttn and SEAD modules.

**Strengths:**

* Innovative Integration: The combination of Mamba and attention mechanisms in the MambaAttn block provides a robust solution for temporal coherence and sequence modeling in gesture generation.
* Comprehensive Multi-Modality Fusion: The SEAD module's ability to disentangle and integrate multiple modalities, including audio, text, style, and emotion, results in more realistic and contextually appropriate gestures.
* Rigorous Evaluation: The extensive experimental evaluation on the BEAT dataset and detailed ablation studies provide strong evidence for the proposed framework's effectiveness.
* Enhanced Performance: The proposed framework outperforms existing methods on key metrics, demonstrating significant improvements in gesture realism and diversity.

**Limitations:**

* Lack of User Study: The paper does not include a user study to evaluate the perceptual quality of the generated gestures. The visual results do not clearly show a distinction from the baseline models, which is a significant omission for a gesture generation task.
* Lack of Experimental Evidence for SEAD: While the SEAD module is claimed to capture emotion and style, there is insufficient experimental evidence to support this claim. The paper lacks specific experiments that demonstrate how SEAD captures these aspects effectively.
* Baseline Implementation Issues: The supplementary material suggests that the baseline model, DiffuseStyleGesture, may not have been properly implemented or optimized, which raises concerns about the validity of the comparative results.
* Objective Metrics Focus: The reliance on objective metrics like FGD, Diversity Score, and BeatAlign, while showing performance improvements, may not fully capture the perceptual quality and effectiveness of the generated gestures. These metrics are not always indicative of real-world performance in generative tasks.
* Code and Writing Similarities: The code and writing style closely resemble those of previous works like MambaTalk and DiffuseStyleGesture, which could indicate that the proposed method is more of an incremental modification rather than a novel contribution.
* Perceptual Quality Concerns: The supplementary videos indicate that the generated gestures do not significantly differ from the baseline models, suggesting that the improvements in objective metrics do not translate to noticeable visual enhancements.

**Suitability:**

3

---

### Official Review · Reviewer_ymtR · 2024-05-24

**Rating:** 4
**Confidence:** 4

**Summary:**

This paper proposes to integrate Mamba architecture into co-speech gestures task. The model combines audio, text, style, and emotion modalities as same as previous method to generate gestures, evaluated on the BEAT dataset, the objective results show a improment in a clear margin.

**Strengths:**

1. sharing the result of integrating Mamba architecture in the context of gesture generation is valuable for community.
2. the desgin of experiments, inputs and output of the pipeline is simple and clear (possitive).
3. objective results are good.

**Limitations:**

As a new baseline paper for using Mamba architecture, I toward to give the positive if the authour solve the following concerns:

1. videos results seems not good. why the gestures seems not continues? like the results from some VQVAEs?
2. will author provide code finally? as a new baseline if the author do not include the codes for implementation, the contribution is limited.

**Suitability:**

3

---

### Official Review · Reviewer_PasF · 2024-05-25

**Rating:** 3
**Confidence:** 3

**Summary:**

The paper introduces MambaGesture, a novel framework integrating MambaAttn with a multi-modality feature fusion module SEAD instead of diffusion models. This is to address the limitations of current approaches in co-speech gesture generation in generating less dynamic and contextually varied gesture by considering a wide range of modalities and analyzing their interactions.

**Strengths:**

- The work, if motivated well, could contribute to contextually relevant co-speech gesture generation.
- The experiments section is well panned and the results presented are  interesting

**Limitations:**

- The problem lacks novelty. Introducing the Mamba-based attention itself is not a sufficient contribution. How it achieves diversity and why existing methods are insufficient to address diversity in gesture generation is to be mentioned. I suggest the authors to better motivate the problem and explain why it is relevant to address this.
-Further, there are multiple works which used multi-modal information apart from audio in co-speech gesture generation.
The argument "Previous research has predominantly focused on audio as the primary input, often neglecting the depth of information available from other modalities." is no longer valid with the existing literature on multimodal co-speech gesture generation. In this scenario, how will you compare your approach with them? Some references are:
 1) Jingyu Wu, Shi Chen, Shuyu Gan, Weijun Li, Changyuan Yang, and Lingyun Sun. 2023. Cultural Self-Adaptive Multimodal Gesture Generation Based on Multiple Culture Gesture Dataset. In Proceedings of the 31st ACM International Conference on Multimedia (MM '23). Association for Computing Machinery, New York, NY, USA, 3538–3549. https://doi.org/10.1145/3581783.3611705
2)Mughal, M. H., Dabral, R., Habibie, I., Donatelli, L., Habermann, M., & Theobalt, C. (2024). ConvoFusion: Multi-Modal Conversational Diffusion for Co-Speech Gesture Synthesis. arXiv preprint arXiv:2403.17936.
3) Ghorbani, S., Ferstl, Y., Holden, D., Troje, N. F., & Carbonneau, M. A. (2023, February). ZeroEGGS: Zero‐shot Example‐based Gesture Generation from Speech. In Computer Graphics Forum (Vol. 42, No. 1, pp. 206-216).
-It would be better to add a subsection on multimodal co-gesture generation
-It will be good if user evaluation is included along with the quantitative analysis.

**Suitability:**

3

---

### Official Review · Reviewer_no2v · 2024-05-28

**Rating:** 3
**Confidence:** 3

**Summary:**

This paper presents "MambaGesture", a novel framework aimed at improving the generation of co-speech gestures using a combination of Mamba-based attention blocks and a multi-modality feature fusion module. The proposed method leverages state space models for enhanced temporal coherence in gesture generation and introduces an innovative feature fusion strategy that incorporates audio, text, style, and emotion.

**Strengths:**

The paper is well-written.

The introduction of the SEA (Style and Emotion Aware) and SEAD (Style and Emotion Aware Disentangled) feature fusion modules is interesting.

**Limitations:**

1. While the integration of the Mamba model is highlighted as a novel contribution, it can be seen as a straightforward replacement of traditional backbones used in similar tasks. This hinders the novelty of the proposed method.

2. The clarity of Table 2 in the paper is questionable. It seems to suggest that the proposed method incrementally adds different components starting from a baseline model BasicDSG+. However, the original BasicDSG+ framework described in the original paper is quite different from the proposed one. Even to verify the effectiveness of Mamba, it is suggested to compare it with a baseline that only differs from the proposed method in the SSM block.

3. It seems that the main improvement is from adding the Mamba, however, since the advantage of Mamba over the Transformer is still under debate (2024), why such a significant improvement is achieved?

**Suitability:**

3

---

### Meta-Review · Area_Chair_Artn · 2024-07-01

**Recommendation:** Accept (Poster)
**Confidence:** 4

**Metareview:**

The paper introduces MambaGesture, a novel framework for co-speech gesture generation that integrates MambaAttn, a Mamba-based attention block, with SEAD, a multi-modality feature fusion module. MambaAttn enhances temporal coherence using sequential data processing and attention mechanisms, while SEAD combines audio, text, style, and emotion modalities through disentanglement, promising more realistic and diverse gestures. Evaluated on the BEAT dataset, the approach achieves state-of-the-art performance in Fréchet Gesture Distance (FGD), diversity scores, and beat alignment. While the paper addresses concerns about novelty, it lacks clarity in presentation and did not provide code during review, promising future release. Objective scores indicate strong semantic relations in generated gestures, validated by a user study assessing motion quality, diversity, and semantic preservation. The authors' commitment to open-sourcing their code enhances transparency, though concerns persist regarding baseline model replication accuracy.